**Data Availability Statement:** All relevant data are within the manuscript and its Supporting Information files.

# A study to assess current approaches of allergists in European countries diagnosing and managing children and adolescents with peanut allergy

Vibha Sharma[1], Jennifer Jobrack[2], Wendy Cerenzia[3], Stephen Tilles[4,5], Robert Ryan [6], Regina Sih-Meynier[4], Stefan Zeitler[7]*, Michael Manning [8]

1 Lydia Becker Institute of Immunology and Inflammation University of Manchester and Royal Manchester Children's Hospital NHS Foundation Trust, Manchester, United Kingdom, 2 Food Allergy Pros LLC, Chicago, IL, United States of America, 3 CE Outcomes, Birmingham, AL, United States of America, 4 Medical Affairs, Aimmune Therapeutics, Brisbane, CA, United States of America, 5 University of Washington, Seattle, WA, United States of America, 6 Aimmune Therapeutics, London, United Kingdom, 7 Aimmune Therapeutics Germany GmbH, Munich, Germany, 8 Medical Research of Arizona, Scottsdale, AZ, United States of America

* szeitler@aimmune.com

## Abstract

### Rationale

Food allergy is documented to result in considerable morbidity, negative impact on quality of life, and substantial medical care costs. Although anecdotal data suggest widely varying practices in the diagnosis and management of food allergies, the diversity and relative frequency of these practices have not been documented.

### Methods

A questionnaire was developed evaluating allergists' management approaches of individuals with peanut allergy (PA) in Germany (DE), France (FR), and the United Kingdom (UK).

### Results

Here, we report the survey results from a total of 109 allergists from DE, FR and the UK. They reported to confirm PA at initial diagnosis using skin prick test ($\geq$60%), while allergists from DE and FR reported using allergen-specific IgE testing more (>86%) compared to the UK (<50%). At initial diagnosis, oral food challenge was used less in DE (13%) and FR (14%) and very rarely in the UK (3%) to confirm diagnosis. Recognition of acute reactions, use of adrenaline auto-injectors and allergen avoidance were reported to be discussed with the patient/caregiver at the initial office visit by most allergists (>75%). Half of the responders reported assessing the patient's quality of life. 63% allergists reported retesting for PA resolution at a later date, with 45% allergists indicated to recommend ingestion of a normal serving of peanut regularly upon resolution. Lack of effective PA treatment was reported to be a 'very significant' barrier for optimal PA treatment, with allergists being less than 'moderately familiar' with data from clinical trials testing new treatments options for PA. Lastly,

**Funding:** This study was supported by Aimmune Therapeutics Inc. in the form of salaries for SZ, RR, ST and RSM, CE Outcomes LLC in the form of a salary for WC, and Food Allergy Pros LLC in the form of a salary for JJ. The specific roles of these authors are articulated in the 'author contributions' section. Editorial support was provided by infill healthcare communication and supported by Aimmune Therapeutics Inc. The funders had no additional role in the study design, data collection and analysis, decision to publish, or preparation of the manuscript.

**Competing interests:** The authors have read the journal's policy and have the following competing interests: SZ, RR, ST and RSM are or were employed by Aimmune Therapeutics Inc. during the course of the study. WC is an employee of CE Outcomes LLC, and JJ is an employee of Food Allergy Pros LLC. VS received speakers' honoraria from Aimmune and is a principal investigator for Aimmune-sponsored ARC008 and ARC005 studies. VS held consultancy briefly with Novartis and Mead-Johnson and received sponsorship to attend and talk at educational events from other pharmaceutical companies. This does not alter our adherence to PLOS ONE policies on sharing data and materials. There are no patents, products in development or marketed products associated with this research to declare.

allergists stated that the severity of patient's PA ranked as the most important factor in their decision to recommend oral immunotherapy for PA treatment.

## Conclusions

This survey provides essential insights into the practice of allergists and highlights some areas that would inform strategies for education and improving PA healthcare.

## Introduction

The prevalence of food allergy (FA), and especially an allergy to peanuts, has increased in developed countries in recent years [1]. In European countries, the prevalence of FA among children aged 6 to 10 years is estimated to be between 1.4% to 3.8% [2].

Unlike other IgE-mediated FAs (allergy to cow's milk or egg) that are often outgrown, peanut allergy (PA) [3] is usually persistent and continues into adulthood in ~80% of affected individuals [4–6]. Peanuts are the most common cause of fatal food-induced anaphylaxis in many European countries [7]. Point prevalence analyses in multiple European countries estimate that 1.6% of European children live with PA, with estimations ranging from 0.24% to 2% depending on the diagnostic methods used [3].

Globally, PA is associated with a considerable disease burden due to multiple psychological, social, and economic factors [8]. A recent Europe-wide study shows that PA has a day-to-day impact on more than 80% of affected children and their parents/caregivers. In comparison, nearly 40% live with a high or extremely high level of stress, and a similar proportion of PA patients reported feeling frequently or very frequently frustrated due to their PA [9].

According to current European guidelines, the standard of care for PA is based on early symptom recognition, short-term symptomatic relief with H1/2 antihistamine blockers, management of acute anaphylactic reactions via intramuscular administration of adrenaline/epinephrine, and long-term strategies to reduce the risk of future reactions by strict avoidance of peanut [10, 11]. At the time of this publication, standardised oral biologic drug–PTAH (Peanut [*Arachis hypogaea*] allergen powder-dnfp) formally called AR101 (brand name PALFOR-ZIA^TM), was approved by the US Food and Drug Administration (January 2020) as the first oral immunotherapy (OIT) indicated to mitigate allergic reactions following accidental exposure to peanuts in individuals aged 4–17 years with a confirmed diagnosis of PA [12], and was not approved by any other regulatory authorities.

Despite their high awareness about the Learning Early About Peanut Allergy (LEAP) study for the early introduction of dietary peanut for high-risk allergic infants [13], and accordingly modified National Institute of Allergy and Infectious Diseases guidelines, a survey reported that many paediatricians continue to have guideline implementation barriers [14].

While attempts are made to improve FA management broadly, to our knowledge, there are no robust analyses that compare the diagnostic and management decisions of allergists in different European countries, specifically in the field of PA. Given the substantial burden on PA patients, caregivers, families and society, the present survey was undertaken to assess existing practices and awareness of allergists caring for young children and teenagers with PA in Germany (DE), France, (FR), the United Kingdom (UK). The data presented in this publication show the survey results from the three European countries only. The ultimate goal of the survey was to document and compare the relative practices of allergists treating PA patients to highlight areas that could be supported by medical education and training in order to improve PA healthcare.

## Methods

### Survey design

A 25-item questionnaire was developed to investigate current allergist approaches to the diagnosis and management of paediatric patients with PA. The survey included two patient case vignettes with associated questions to assess management choices, knowledge, attitudes and perceived barriers to optimal care for patients with PA. The survey was translated for FR and DE allergists, and distributed via email to allergists in FR, DE, UK and the United States of America during July 2019. The responses were collected anonymously, and data from FR, DE and UK are presented in the current manuscript. Detailed descriptions of case vignettes and questions used in the survey are available in the Supplementary Information.

### Participant recruitment

The survey was distributed to 6450 allergists in three European countries (1185 in FR, 3350 in DE, 1915 in the UK). The inclusion criterion was physicians actively managing patients younger than 18 years of age with PA. A monetary incentive (equivalent to 50 USD) was offered to allergists for their participation, which was expected to take about 25 minutes to complete.

### Statistical analysis

Survey responses were collected via an online platform. Survey data were compiled from each country, and open-ended responses from FR and DE were translated to English. The data were analysed with IBM SPSS Statistics 25. Descriptive statistics (frequencies and means) were calculated to examine overall responses and related trends among survey items. Inferential statistics (analysis of variance (ANOVA) for the Likert scale questions, and Chi-square distribution test for select one or select all questions) were performed to interpret differences between allergists responses, practising in different countries. Differences between groups were considered statistically significant at p $\leq$0.05.

## Results

### Study participants

A total of 109 allergists (38 from DE, 36 from FR and 35 from the UK) completed the survey, with a response rate of 1.69% for all countries (1.13% for DE, 3.04% for FR and 1.83% for the UK). The allergists enrolled in this survey reported seeing a similar number of patients with PA per month, with an average of 14 patients per month (Table 1). Most patients who attended

**Table 1. Demographics of physicians who participated in the survey.**

|  | Average* (n = 109) | DE (n = 38) | FR (n = 36) | UK (n = 35) |
|---|---|---|---|---|
| Mean number of patients with PA seen per month | 14 | 15 | 12 | 16 |
| Mean number of PA patients <18 years seen per month | 10 | 9 | 9 | 13 |
| Primary practice setting (%) |  |  |  |  |
| Private/Community | 65 | 71 | 89 | 34 |
| Academic/University | 35 | 29 | 11 | 66 |
| Practice location (%) |  |  |  |  |
| Urban | 77 | 74 | 78 | 80 |
| Suburban | 13 | 16 | 8 | 14 |
| Rural | 10 | 11 | 14 | 6 |

Data regarding the practice characteristics were collected on each respondent as a part of the survey. PA, peanut allergy.

*Represents the average value of France (FR), Germany (DE) and the United Kingdom (UK).

were less than 18 years of age in all countries, with an average of 10 patients per month. In DE and FR, the majority of allergists' primary practice was private or community-based, while in the UK, most allergists practised in an academic or university setting.

**Primary assessment approaches for diagnosis and management of PA diagnosing and verifying PA.** At initial diagnosis, 71% of allergists (61% DE, 86% FR, 66% UK) reported performing a skin prick test (SPT) and 77% (95% DE, 86% FR, 49% UK) reported performing an allergen-specific IgE (sIgE) test for a 2-year old child's first PA evaluation (Fig 1A).

In an adolescent patient with long-standing PA and difficulty adhering to peanut avoidance, 64% of allergists (61% DE, 72% FR, 60% UK) reported performing a SPT, and 63% (66% DE, 78% FR, 46% UK) reported performing a sIgE test to reconfirm a PA diagnosis (Fig 1A). At that same point, 32% of allergists (37% DE, 42% FR, 17% UK) would choose to perform an oral food challenge (OFC) to reconfirm the PA diagnosis for the adolescent patient (S1 Table Q6). In a scenario where the patient's sIgE and SPT results were indeterminate at retesting, almost half of all respondents in each country (45% DE, 47% FR, 43% UK) would perform an OFC (S1 Fig in S1 File). In contrast, 39% of allergists (39% DE, 29% FR, 49% UK) would recommend their patient to continue peanut avoidance without further testing. At this stage, 13% of allergists (13% DE, 22% FR, 3% UK) would recommend OIT to the patient (S1 Table Q8). 14% of allergists (21% DE, 19% FR, 3% UK) reported never conducting OFCs in their patients (S1 Table Q9).

**Initial management.** At the initial office visit of a child with a potential PA diagnosis, 92% of allergists (92% DE, 97% FR, 86% UK) reported discussing recognition of acute reactions personally with the patient's caregiver (Fig 1B). At this stage, 79% of allergists (79% DE, 81% FR, 77% UK) stated discussing the usage of epinephrine/adrenaline auto-injector, whereas 91% (92% DE, 94% FR, 86% UK) reported discussing allergen avoidance personally with their patient's caregiver (Fig 1B).

In a patient with long-standing PA returning with difficulty avoiding peanuts, 65% of allergists (66% DE, 58% FR, 71% UK) reported to renew or revise the emergency action plan of the patient (Fig 1B). Additionally, 64% of allergists (58% DE, 50% FR, 83% UK) would reinforce previous education about PA management, and 34% (42% DE, 33% FR, 26% UK) would assess patient's nutritional status (Fig 1B).

At the initial office visit of a child with potential PA, 62% of allergists (61% DE, 61% FR, 63% UK) reported discussing the impact of PA on Quality of Life (QoL) personally with the child's caregiver (Fig 1C). As part of the initial management of a long-standing PA patient returning with avoidance difficulties, 50% of allergists (42% DE, 47% FR, 60% UK) reported performing QoL assessment (Fig 1C) wherein the majority (81% DE, 82% FR, 81% UK) opted for subjective QoL assessment, rather than using a standardised assessment tool (S2 Fig in S1 File). For the same patient, on a scale from 'not at all' (1) to 'extremely significant' (5), allergists responded that maximising patient's QoL, preventing serious reactions and relieving parent anxiety were between 'very' and 'extremely significant' goals in managing that patient (S3 Fig in S1 File).

In general, when asked to rate on the level of significance, how does PA negatively affect QoL of their patients, from 'not at all' (1) to 'extremely significant' (5), respondents from all countries stated 'moderately significant' (3.66 DE, 3.56 FR, 3.54 UK, S3 Fig in S1 File).

**Retesting and follow-up management.** After initial management of a child with newly diagnosed PA, 63% of allergists (58% DE, 61% FR, 69% UK) reported retesting the resolution of PA at a later date (Fig 1D). With regards to retesting frequency for the same patient, 29% of allergists (18% DE, 56% FR, 14% UK) stated to follow patients up bi-annually, whereas 24% (5% DE, 22% FR, 46% UK) would see the patient yearly (S4 Fig in S1 File). A majority of DE allergists (53%) would follow-up with the patient every three months (S1 Table Q3). In

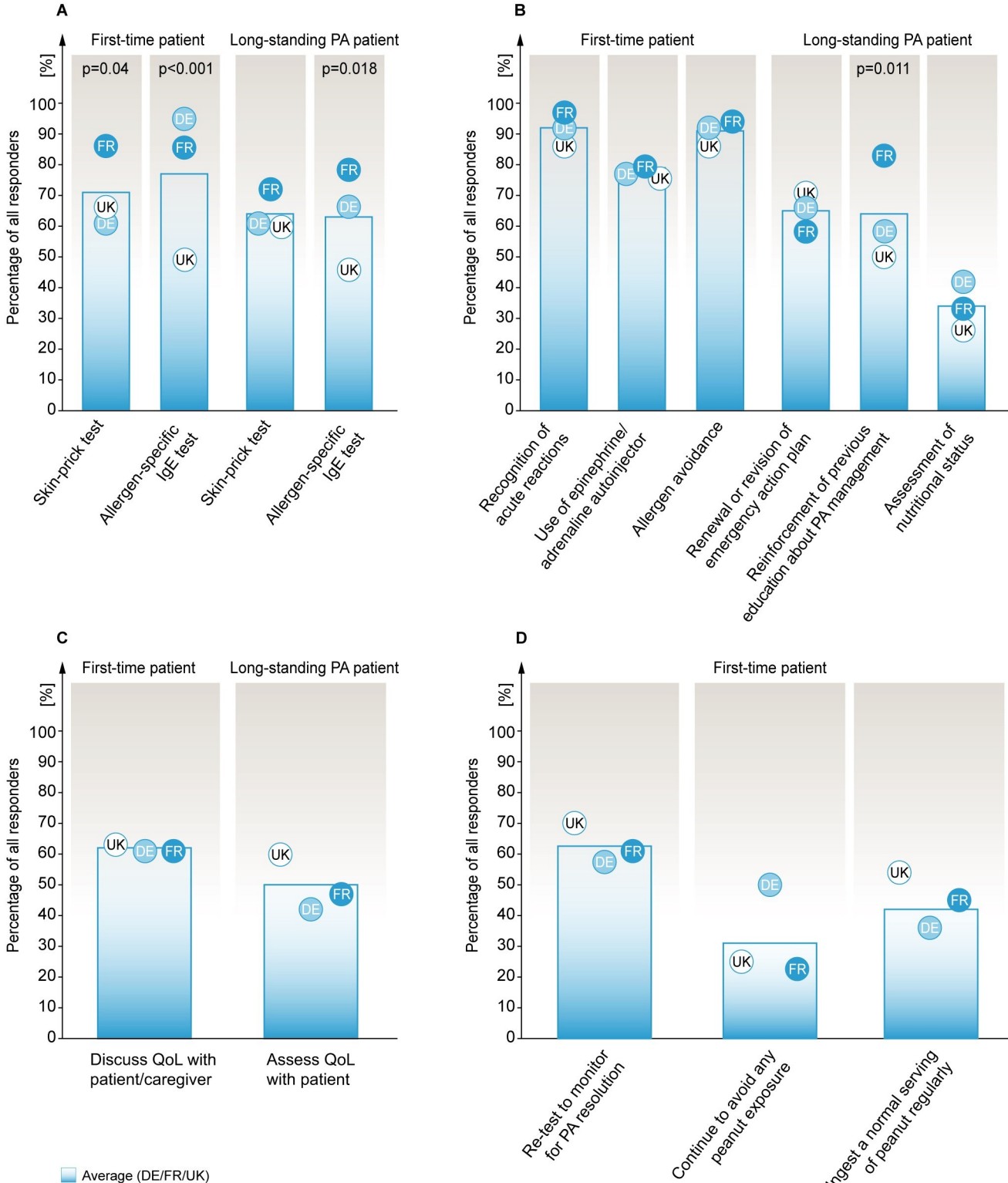

**Fig 1. Primary assessment approaches towards PA diagnosis and management. Panels A-D**: Percentage of all responses are shown. **A.** Choices of diagnostic tests performed to confirm PA for the first-time patient with potential PA and a long-standing patient for re-confirming PA. **B.** Topics discussed by allergists at the initial office visit with a first-time or long-standing patient with PA and their caregivers. **C.** Discussions about QoL with patients/caregivers. **D.** Monitoring for resolution of PA after confirming the diagnosis in a first-time PA patient. Blue gradient bar shows average results for responses from France (FR), Germany (DE) and the United Kingdom (UK) (n = 109; FR (36), DE (38), UK (35)), and circles represent individual

countries with the respective two-letter acronym. Only statistically significant p values from Chi-Square distribution analyses comparing differences among the three individual countries. (p ≤ 0.05). IgE, Immunoglobulin E; PA, peanut allergy; OIT, oral immunotherapy; OFC, oral food challenge.

addition, 10% of allergists (13% DE, 3% FR, 14% UK) stated to follow-up only as needed (S4 Fig in S1 File).

Upon the patient's PA resolution at a later date, 45% of allergists (36% DE, 45% FR, 54% UK) stated to recommend the patient ingest a normal serving of peanut regularly (Fig 1D). In comparison, 33% of allergists (50% DE, 23% FR, 25% UK) would recommend the patient to continue complete avoidance of peanut exposure (Fig 1D).

## Factors influencing decision making regarding PA treatment

Out of five presented issues, allergists rated lack of effective PA treatments other than avoidance, the ubiquity of peanut in the patients' environment, and misconceptions or myths about PA as 'moderate' (≥ 3) to 'very significant' (≥ 4) barriers affecting optimal management of their PA patients (Fig 2A, S1 Table Q13).

In response to rating their familiarity of four emerging therapies for PA (PTAH/AR101 OIT, peanut subcutaneous immunotherapy (SCIT), peanut sublingual immunotherapy (SLIT) and peanut epicutaneous immunotherapy (EPIT)), allergists from all countries responded to be between 'slightly' (≥ 2) and 'moderately familiar' (≥ 3) for all four emerging therapies, except DE and UK allergists being less than 'slightly familiar' with EPIT (1.97) and SCIT (1.91) respectively (Fig 2B, S1 Table Q19).

Around half of all allergists (37% DE, 42% FR, 54% UK) responded correctly to which among the three routes of immunotherapies (SLIT, EPIT or OIT) demonstrated a 100-fold increase from baseline in the median tolerated dose after 12 months of treatment in a well-controlled clinical trial setting, with OIT being the correct option (S1 Table Q17). Around half of all allergists were 'unaware/unsure' about the safety data from different clinical trials comparing OIT with two routes of immunotherapies (SLIT or EPIT) as treatments for PA (S1 Table, Q18). Also, around 34% of allergists (34% DE, 39% FR, 29% UK) were aware that EPIT as PA treatment was not associated with similar rates of adverse reactions from different clinical trials compared to OIT (S1 Table Q18).

Allergists from all countries stated being between 'moderately' (≥ 3) and 'very concerned' (≥ 4) about the following aspects related to use of investigational OIT for PA treatment: risk of adverse effects, the need for maintenance dosing, lack of efficacy, patient's reluctance to undergo immunotherapy, patient's lack of adherence to immunotherapy treatment, lack of data supporting long-term outcomes and logistics of therapy administration (Fig 2C and S1 Table Q23).

Allergists were also in agreement that data supporting efficacy and safety profiles were between 'very' (≥ 4) and 'extremely significant' (5) factors in selecting between treatments, if multiple immunotherapies became available for PA treatment (Fig 2D, S1 Table Q21). Also, other factors such as the potential for loss of desensitisation when therapy is discontinued, the burden of scheduling and time required for treatments, cost or insurance coverage, convenience, and ability to assess patient response were between 'moderate' (≥ 3) and 'very significant' decision elements when choosing between multiple immunotherapies (Fig 2D and S1 Table Q21). Allergists also stated that efficacy and safety data from clinical trials, real-world data, approval by The Food and Drug Administration and inclusion of treatment in nationally recognised treatment guidelines were between 'very' (≥ 4) and 'extremely important' (5) factors in improving their comfort level in considering implementation of a new drug or treatment (S1 Table Q22).

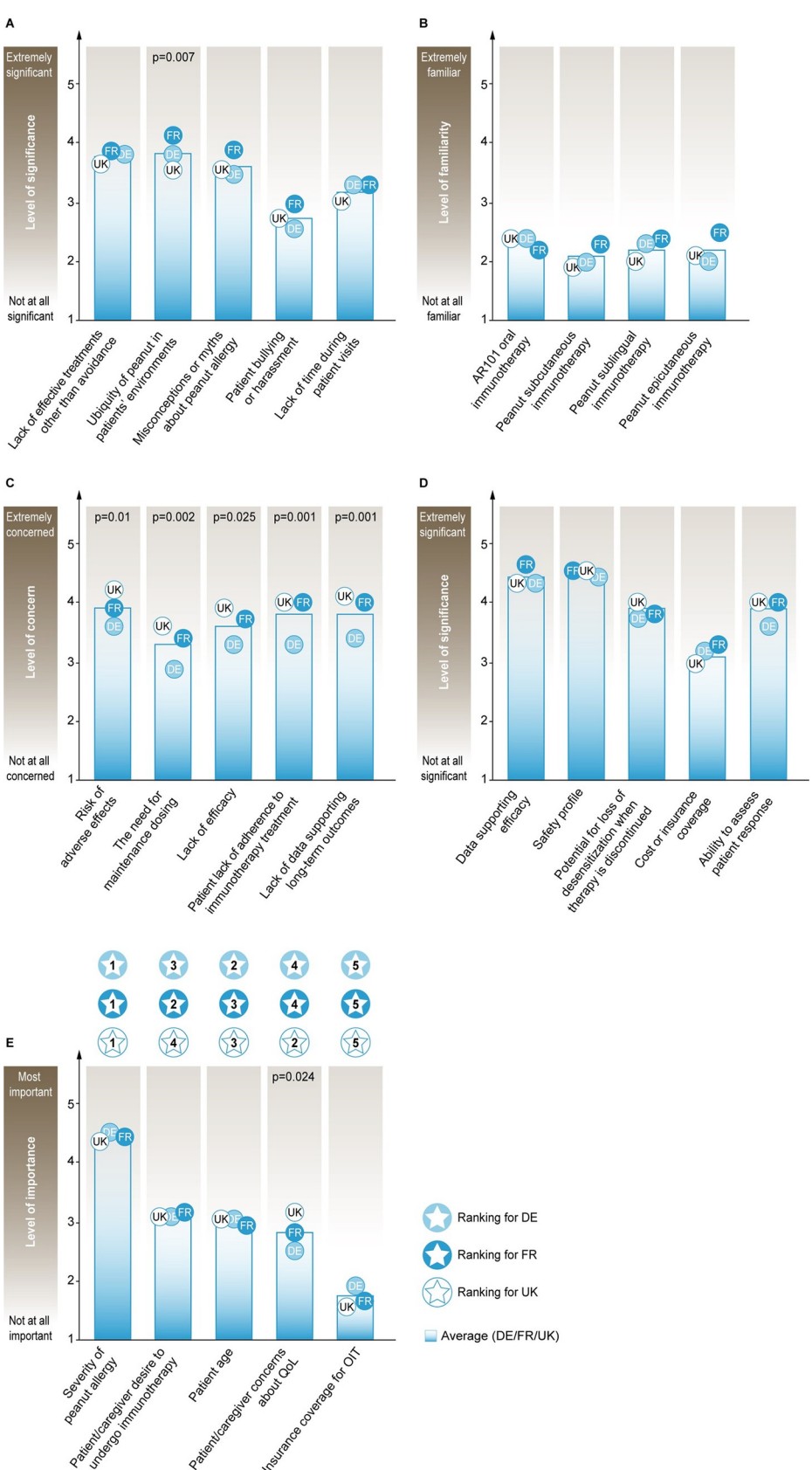

**Fig 2. Factors influencing decision-making regarding PA treatment. Panels A-E:** Mean scores of allergists' responses are shown. **A.** Significant barriers to the optimal management of patients with PA. **B.** Familiarity with clinical trials regarding emerging therapies for PA. **C.** Concerns with aspects related to investigational OIT for PA. **D.** Significance of factors in selecting between treatments, if multiple immunotherapies for PA become available. **E.** Factors important in the decision to recommend OIT for PA treatment ranked with the highest score at 1st Rank and lowest at 5th Rank; Blue gradient bar shows average results for responses from France (FR), Germany (DE) and the United Kingdom (UK) (n = 109; FR (36), DE (38), UK (35)), and circles represent individual countries with the respective two-letter acronym. Only statistically significant p-values from ANOVA tests comparing differences among the three individual countries ($p \leq 0.05$) are shown in figures. PA, peanut allergy; OIT, oral immunotherapy.

When considering OIT for a patient with PA, allergists were asked to rank five presented factors on a scale of importance (1 = most important and 5 = least important). The severity of PA of the given patient was ranked as the most important factor by allergists from all countries. Factors such as patient' s/caregiver's desire to undergo OIT, patient's/caregiver's concerns about QoL and patient's age were ranked second, third and fourth important factors. However, their order was different for the three surveyed countries (Fig 2E, S1 Table Q20). Insurance coverage for OIT ranked as the least important factor for all allergists while considering OIT for their PA patient (Fig 2E, S1 Table Q20).

## Multidisciplinary care and shared decision-making for PA management

With regards to questions involving multidisciplinary care, two-thirds of all allergists (63% DE, 61% FR, 77% UK) reported typically including other HCPs in the management of their PA patients, with a dietician or a nutritionist as the preferred choice by allergists (92% DE, 82% FR, 96% UK) to refer their patients for PA management (S1 Table Q10 and Q10a).

In response to questions involving patient/caregiver in the final decision-making for their PA management, almost half of all allergists (29% DE, 58% FR, 46% UK) opted to involve the patient/parent in the final treatment decision-making (S1 Table Q16). Besides, one in three DE allergists (32%) would instead decide the treatment for their PA patients on their own, compared to around one in five in FR (19%) and one in ten in the UK (11%) (S1 Table Q16).

## Discussion

Given the significant burden associated with PA, improving care for PA patients (especially children and adolescents) and caregivers remains a challenge for allergists globally. Identifying areas of support around allergists' education and training is critical for improved patient care and management of PA. This international survey provides essential insights regarding practising allergists' awareness and knowledge related to PA treatment in developed countries.

As PA is an IgE-mediated type I hypersensitivity reaction of the immune system to the consumption of or exposure to peanuts [15, 16], current European guidelines (EACCI) state that clinical diagnosis of PA can be made by the combination of a clinical presentation and evidence of peanut-specific IgE detected either by a positive SPT or sIgE test [17]. Results from the current study show that allergists commonly used diagnostic tests to confirm PA were SPT and sIgE test from all surveyed countries. Still, some reported performing either of them in the absence of a clear clinical presentation (S1 Table Q1). Of note, more than one-third of all respondents stated they would choose not to do a SPT in a returning patient with a history of PA. SPT and sIgE are sensitive methods to confirm the diagnosis of most FAs with positive predictive values generated based on the probability of allergic reaction during an OFC [18].

OFC may occasionally be required to make a definitive diagnosis of a FA. Still, with the advent of more specific diagnostic methods (such as allergen component testing and basophil activation test), the level of risk involved and the resource-intensive nature of conducting an

OFC calls for its use only for equivocal cases [19]. In consensus with the FA practice parameter update [20], if the patient has a convincing clinical history of an allergic response to the allergen, then performing OFC is deemed as high-risk and should ideally not be performed [21]. However, in case initial or follow-up diagnostic tests are negative or inconclusive (as presented in the survey, S1 Table Q8), then conducting an OFC is deemed low-risk and crucial, as a negative OFC result would contribute to a significant improvement in patient's QoL [21].

The survey results also show that most allergists across all surveyed countries reported they would discuss how to recognise acute reactions and avoid allergens at the initial visit with their patient/caregiver (Fig 1B). The British Society for Allergy and Clinical Immunology (BSACI) guidelines state that the safe management of anaphylaxis depends on early recognition and rapid intervention with epinephrine/adrenaline [22]. Renewal/revision of an emergency action plan, reinforcement of previous education about PA management, and discussing effects of PA on the QoL of patient/caregiver were other initial management steps reported to be performed by >40% allergists. An earlier study demonstrated that even though 95% of allergists (n = 500) adhered to the use of practice recommendations for the treatment and management of anaphylaxis, opportunities for improvement of patient education and information sharing were found, including providing of emergency plans and revising pre-existing plans [23], which is in line with this survey's findings.

Considering that the natural course of PA is different from other FAs, and does not resolve during early childhood [24], the current survey also provided insights on allergists' decisions regarding the frequency of follow-up evaluations to check for patient's PA resolution. Almost half of all allergists would not or were unsure of retesting their patients for resolution of PA at a later date (Fig 1D), a result that in light of other publications signifies potential impact on the QoL of the patient/caregiver due to psychological, social and economic burden on the affected family and society, caused by fear of accidental peanut exposure, bullying or feeling of being left out from social groups and events and maintenance of a peanut avoidance diet [9, 25]. In patients with resolved PA, half of the surveyed allergists indicated they would suggest the patient to consume a normal peanut intake regularly, while the other half would recommend continuing strict avoidance of peanut for their patients despite resolved PA (Fig 1D).

Allergists in the survey rated lack of effective treatments other than avoidance as a 'moderate' to 'very significant' barrier to PA management (Fig 2A), which was expected considering this survey was carried out before the first US approval of PTAH/AR101/ [26] and the lack of an approved PA treatment in Europe at present. The current investigational treatment landscape for PA includes immunotherapy administered via oral, sublingual, subcutaneous or epicutaneous routes [27]. A series of questions posed to the allergists in the survey showed that almost half of allergists were unsure/unaware of efficacy and safety data from different clinical trials regarding emerging therapies for PA (S1 Table Q17 & Q18). Allergists indicated that efficacy and safety data from clinical trials were between 'very' to 'extremely important' factors when recommending a new drug or treatment to their patients (S1 Table Q22). Currently, OIT efficacy is the highest compared to other routes for PA treatment, with data from large phase 3 trials concluding that treatment with PTAH AR101/OIT resulted in desensitisation in children (4–17 years) who were highly allergic to peanut [28]. Despite its clear efficacy profiles in terms of desensitisation, OIT is also associated with higher rates of systemic adverse events [27] which have recently been analysed in detail by two large meta-analyses [29, 30]. These observations are consistent with the proposal that physicians, prospective patients and their caregivers should approach OIT as a nuanced and shared decision. Potential patients and caregivers should weigh the risks of treatment-related side-effects of OIT against the uncertainty and risk of allergic reactions due to accidental exposure inherent with strict avoidance only [31].

The current survey also highlights some of the ambiguity regarding OIT among the allergist community. Allergists were generally 'very concerned' about the safety and long-term OIT as PA treatment (Fig 2C, S1 Table Q23). With regards to choosing the appropriate treatment for their PA patient, in a scenario where multiple therapies would be approved, allergists were in unison that data supporting treatment efficacy, safety and post-treatment loss of desensitisation were between 'moderate' to 'very significant' factors (Fig 2D). These results indicate the need for better communication of ongoing and future studies with promising efficacy and safety profiles that could improve PA management.

Data from the current survey show most allergists would typically include other HCPs in the multidisciplinary management of their PA patients (S1 Table Q10). These findings are in line with the previous results from US allergists, who reported that 20% of patients with anaphylaxis referred to primary care and emergency department physicians are misdiagnosed [32]. Also, when considering shared decision-making regarding PA treatment with their patients/caregivers, most allergists stated their preference to share or partially share the responsibility for deciding the optimal treatment (S1 Table Q16). To highlight the importance of multidisciplinary management of PA, recent reports highlight the continuous efforts made to help PA caregivers navigate through emerging therapies and decide on the best treatment choice for their patients [33, 34].

Some limitations should be considered when interpreting the results of this study. Heterogeneity among groups concerning allergists' qualifications, practice settings and locations leading to possible variations in levels of experience, knowledge and awareness compared to the norm. Selected countries were targeted to represent practices in PA management in developed countries. Although the survey was distributed to a large number of allergists in all countries, the sample size of 109 allergists who completed the survey suggests that the results should be interpreted with care, as the number of respondents may not be fully representative. One could speculate that the complexity of the questions and the amount of time taken to complete the survey (around 25 minutes) could be a reason for the low response rate. Additionally, the focus of the questions and the choice of the two case vignettes could have led to answer bias and does not represent the complete spectrum of PA cases that exist.

## Conclusion

This study revealed essential insights into the current allergists' approaches to the diagnosis and management of their PA patients. Allergists from surveyed countries reported the use of different diagnosis and management methods in treating PA patients, including confirmation of PA diagnosis, performing OFCs, discussions of initial management issues or follow-ups to check for PA resolution. Further, it also demonstrates the allergists' self-reported levels of awareness of and concerns around the data from emerging PA therapies and the use of multidisciplinary collaboration and shared decision making with the patient/caregivers.

## Supporting information

**S1 File.**
(TIF)

**S1 Table. Supplementary results.**
(DOCX)

**S2 Table. A list of all survey questions.**
(DOCX)

## Acknowledgments

This study was sponsored by Aimmune Therapeutics. The authors would like to thank Sylvie Stacy, Andrea Vereda, and Ian Hitchcock for valuable input into the study design, Lee Baylis for critical review of the manuscript, and Brandon Coleman for providing analytical support for this study. Editorial assistance and medical writing support were offered by infill healthcare communications and funded by Aimmune Therapeutics.

## Author Contributions

**Conceptualization:** Wendy Cerenzia, Stephen Tilles, Regina Sih-Meynier.

**Investigation:** Wendy Cerenzia, Regina Sih-Meynier.

**Methodology:** Wendy Cerenzia, Regina Sih-Meynier.

**Project administration:** Wendy Cerenzia, Regina Sih-Meynier.

**Resources:** Robert Ryan, Regina Sih-Meynier, Stefan Zeitler.

**Supervision:** Vibha Sharma, Jennifer Jobrack, Wendy Cerenzia, Robert Ryan, Regina Sih-Meynier, Stefan Zeitler, Michael Manning.

**Writing – original draft:** Vibha Sharma, Jennifer Jobrack, Wendy Cerenzia, Stephen Tilles, Robert Ryan, Regina Sih-Meynier, Stefan Zeitler, Michael Manning.

**Writing – review & editing:** Vibha Sharma, Jennifer Jobrack, Wendy Cerenzia, Stephen Tilles, Robert Ryan, Regina Sih-Meynier, Stefan Zeitler, Michael Manning.

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
