## [Decision Letter · Decision Letter 0]

19 Oct 2020

A study to assess current approaches of allergists in European countries diagnosing and managing children and adolescents with peanut allergy

PONE-D-20-27455

Dear Dr. Ryan,

We’re pleased to inform you that your manuscript has been judged scientifically suitable for publication and will be formally accepted for publication once it meets all outstanding technical requirements.

Kind regards,

Davor Plavec, MD, MSc, PhD, Prof.

Academic Editor

PLOS ONE

Journal Requirements:

1. Please include additional information regarding the survey or questionnaire used in the study and ensure that you have provided sufficient details that others could replicate the analyses. For instance, if you developed a questionnaire as part of this study and it is not under a copyright more restrictive than CC-BY, please include a copy, in both the original language and English, as Supporting Information.

2. In the methods section, please clarify whether or not the online survey data was collected anonymously.

3.Thank you for stating the following in the Financial Disclosure section:

[The study was supported by Aimmune Therapeutics Inc. The author(s) received no specific funding for work on this publication. The funders had no role in study design, data collection and analysis or preparation of the manuscript. Editorial support was provided by infill healthcare communication and supported by Aimmune Therapeutics Inc.].   

We note that one or more of the authors are employed by a commercial company: Food Allergy Pros LLC, CE Outcomes

Please respond by return email with an updated Funding Statement and Competing Interests Statement and we will change the online submission form on your behalf.

Reviewers' comments:

Reviewer's Responses to Questions

**Comments to the Author**

1. Is the manuscript technically sound, and do the data support the conclusions?

Reviewer #1: Yes

Reviewer #2: Yes

2. Has the statistical analysis been performed appropriately and rigorously? 

Reviewer #1: Yes

Reviewer #2: Yes

3. Have the authors made all data underlying the findings in their manuscript fully available?

Reviewer #1: Yes

Reviewer #2: Yes

4. Is the manuscript presented in an intelligible fashion and written in standard English?

Reviewer #1: Yes

Reviewer #2: Yes

5. Review Comments to the Author

Reviewer #1: In the manuscript entitled “A study to assess current approaches of allergists in European countries diagnosing and managing children and adolescents with peanut allergy” the authors evaluated a questionnaire completed by allergists in Germany, France and the UK.Unfortunately the suvey was completed by only 109 allergists in the countries tested, or 1.69%, which is actually a very low response rate.

Therefore, the authors point out well that this fact should be taken into account when interpreting the results. An additional limiting factor is heterogeneity between groups.

Recommendation to the authors: to make additional analysis why the response rate was so low, and to conduct extended research in other developed EU countries.

Reviewer #2: This study revealed essential insights into the current allergists’ approaches to the diagnosis and management of their PA patients. Allergists from surveyed countries reported the use of different diagnosis and management methods in treating PA patients, including confirmation of PA diagnosis, performing OFCs, discussions of initial management issues or follow-ups to check for PA resolution.

6. PLOS authors have the option to publish the peer review history of their article (what does this mean?). If published, this will include your full peer review and any attached files.

Reviewer #1: No

Reviewer #2: No

---

## [Editor Report · Acceptance letter]

20 Nov 2020

PONE-D-20-27455 

A study to assess current approaches of allergists in European countries diagnosing and managing children and adolescents with peanut allergy 

Dear Dr. Ryan:

I'm pleased to inform you that your manuscript has been deemed suitable for publication in PLOS ONE. Congratulations! Your manuscript is now with our production department. 

Kind regards, 

on behalf of

Dr. Davor Plavec 

Academic Editor

PLOS ONE